# Problems with Congestive Heart Failure and Lameness That Have Increased in Grain-Fed Steers and Heifers

**DOI:** 10.3390/ani14192824

**Published:** 2024-09-30

**Authors:** Temple Grandin

**Affiliations:** Department of Animal Science, Colorado State University, Fort Collins, CO 80523, USA; cheryl.miller@colostate.edu; Tel.: +1-970-310-5411

**Keywords:** welfare, cattle, lameness, genetic selection, congestive heart failure, feedlot

## Abstract

**Simple Summary:**

There are increasing problems with both lameness and congestive heart failure in heavy, grain-fed steers and heifers. An inspection of hearts at the slaughter plant indicated that increasing numbers of these cattle have abnormally swollen hearts. This may increase death losses late in the feeding period. Lameness and hoof abnormalities in fed cattle have also increased. There is a possible relationship between both of these conditions and genetic selection for economically important meat traits. Both conditions seriously compromise animal welfare.

**Abstract:**

Grain-fed steers and heifers have increasing problems with both lameness and congestive heart failure. Congestive heart failure used to be limited to cattle raised at high altitudes. It is now occurring at much lower elevations. An inspection of hearts at the slaughter plant indicated that some groups of grain-fed steers and heifers had abnormally swollen hearts in 34% of the animals. Congestive heart failure may also increase death losses in the late stages of the feeding program. Lameness has also increased to 8% of grain-fed steers and heifers arriving at U.S. slaughter plants. Twenty years ago, observations by the author indicated that lameness in fed cattle arriving at the slaughter plant was almost zero. There has also been an increasing occurrence of abnormal hoof structure and poor leg conformation. Genetic selection for rapid growth and a large ribeye may be associated with the increase in both of these problems. Lameness and heart problems severely compromise animal welfare.

## 1. Introduction

Two problems that severely compromise cattle welfare are now occurring more frequently in heavy, grain-fed steers and heifers. Approximately 8% of the cattle arriving at U.S. slaughter plants had some degree of lameness [1]. According to a survey conducted in 2022 at one large fed beef plant, 32% of the heavy, grain-fed steers and heifers were lame [2]. In the first survey 7.8% of the cattle had minor stiffness and they kept up with other cattle at a walk [1]. Less than 1% were more severely lame. During the last ten years, there have also been increasing problems with congestive heart failure [3,4,5]. It increases death losses in heavy, grain-fed steers and heifers later in the feeding period [3]. I have observed cattle handling for over 50 years. Twenty years ago, lameness and congestive heart failure was almost never a problem in the cattle arriving at the slaughter plant. My occupation is a university professor and livestock industry consultant. The main focus of my work is the design of cattle handling systems and the implementation of welfare assessments [6,7,8,9]. My work provided me with the opportunity to observe thousands of cattle being moved in slaughter plants. Until the mid-2000s, the occurrence of lameness that made grain-fed steers and heifers difficult to move and handle at the slaughter plant was almost zero. The cattle moved easily and were willing to walk. The main problem during the first half of my career, during the 1970s and 1990s, was poor, rough handling practices. I worked on designing more efficient cattle equipment facilities and I also developed a simple scoring system for assessing cattle handling at slaughter plants [10]. Cattle-handling practices in both plants and feedlots were really rough and an electric prod was used on almost every animal [8]. In the 1970s, I visited the Swift beef slaughter plant in Tolleson, Arizona, every week for three years. I observed thousands of grain-fed steers and heifers at this plant and they moved easily. At this time, it never occurred to me that lameness in these cattle would become an issue in the future. Twenty years ago, there were no studies on the percentage of lame, grain-fed steers and heifers arriving at the slaughter plant. In 1999, large meat-buying companies started auditing animal welfare in the slaughter plants that supplied them. I was hired to train their food safety auditors to conduct welfare audits [7,8]. When large companies, such as the McDonald’s Corporation, started inspecting their suppliers, handling practices quickly improved. Handling practices were scored for electric prod use, falling during handling, and cattle vocalization [7,10]. Within six months, electric prod use, falling, and vocalization in the stunning area were greatly reduced [7,8]. The penalty for not improving was being removed from the buyer’s approved supplier list. Between 1999 and 2005, I trained animal welfare auditors in numerous fed beef slaughter plants in Colorado, Nebraska, Kansas, and Texas. This provided me with the opportunity to observe thousands of grain-fed steers and heifers being moved through pens, alleys, and races. Both handling and stunning practices were greatly improved in the majority of plants. Plant managers increased employee training on low-stress handling methods, provided better supervision, repaired broken equipment, and installed non-slip flooring in high traffic areas [7].

Low stress, quiet cattle-handling practices provided many advantages, such as improved beef quality and less bruising. Poor handling practices, such as excessive electric prod use, may make beef tougher and increase bruising [11,12]. The U.S. beef industry has recognized the value of low-stress handling. They now have good training materials for instructing the people who handle the cattle [13,14]. Research clearly shows that stockmanship training improves cattle handling [15,16]. To maintain high animal welfare standards, it is essential to have cattle that will move easily with very low levels of lameness. When cattle refuse to move, they are more likely to be subjected to stressful, aversive handling practices [16,17]. Refusing to move forward may also be caused by visual distractions, such as sharp shadows or a noisy truck [17]. In the next section of this paper, I will review research on both congestive heart failure and lameness. Both of these problems may make moving heavy, grain-fed steers and heifers more difficult. Research studies have also shown that both problems have recently increased.

## 2. Increasing Congestive Heart Failure in Grain-Fed Steers and Heifers

This non-infectious condition was originally called Brisket disease because fluid buildup would cause swelling of the brisket [18]. This is one sign that the heart is starting to fail [19]. Congestive heart failure also has other symptoms, such as weakness and exercise intolerance. Pulmonary arterial pressure (PAP) is the resistance to blood flow through the lungs [20]. Cattle with a higher resistance are more likely to get brisket disease [21]. Cattle can be tested to determine the level of blood flow resistance through the pulmonary artery [21]. This is now being used to predict their susceptibility to congestive heart failure. In the 1960s, brisket disease was usually limited to high altitudes between 2438 m and 3657 m [22]. Another name for it was “mountain sickness” [22]. In 2010, the Canadian researcher Sebastian Buczinski [23] stated that heart disease in cattle had a low incidence [23]. Some of the early indications that congestive heart failure was occurring at lower altitudes was research conducted from 2015 to 2018 [20,24,25]. Between the years of 2000 and 2012, the necropsies of dead cattle at feedlots indicated that the prevalence of congestive heart failure had doubled [25]. Another study indicated that heart problems were starting to occur at moderate elevations of 1600 m [26]. Heart issues became more obvious when cattle feeders started reporting that they were having more cattle that died shortly before slaughter [26]. Initially, many feedlot managers thought the cattle had bovine respiratory disease. Treating them with antibiotics had no effect. Heart problems were being misdiagnosed [3].

### Genetic Factors May Be Associated with the Increase in Congestive Heart Failure

Researchers began to suspect that genetics were associated with the increased cases of bovine congestive heart failure. The earlier research studies were based on necropsy reports on dead cattle at feedlots [25]. The seriousness of the problem was identified when researchers started evaluating hearts at the slaughter plants. A visual scorecard was used to evaluate the hearts for differing amounts of swelling. Scoring tools that show normal beef hearts with different amounts of swelling are available [3,4]. Both of these papers are Open Access.

In 2021, researchers used a five-point scoring system at the slaughter plant [4]. Thirty-four percent of the finished cattle from a Colorado feedlot exhibited signs of heart swelling [4]. The authors suggested that it may be related to the intensive genetic selection of beef cattle [4]. By 2022, researchers discovered that genetic factors were associated with congestive heart failure. In some groups of feedlot cattle there was 7% mortality [3]. Death losses increased with more days on feed [3]. Animals with two copies of at-risk alleles were 28-fold more likely to have congestive heart failure [3]. One of the most recent studies, in 2023, showed that 4% of Angus cattle had signs of late-stage congestive heart failure [5]. The hearts were scored at slaughter and the elevation was only 756 m. The feedlot was located in Idaho. They also found genetic correlations with the meat traits of high economic importance [5]. An earlier study showed that high pulmonary artery (PAP) scores were associated with growth [20]. Another study showed that selection for a large ribeye may result in more heart problems [27]. The thickness of the backfat and marbling had almost no relationship [27]. The probability of death from heart failure increased with increasing days on feed, until 326 days [28]. The research clearly shows that problems with congestive heart failure have greatly increased.

In 2022, a feedlot manager in Colorado informed me that that he had traced cattle that died late in the feeding period back to a single Angus sire. This feedlot was at an altitude of 1400 m. A research study also showed that there was a sire effect on the heart score at slaughter [4]. The feedlot manager was able to determine the sire because they raised beef-on-dairy steers. Holstein cows on a dairy farm were bred by artificial insemination to Angus bulls to produce beef feedlot steers. This is becoming an increasingly popular practice [29]. They had very good records, which made it easy to determine the sire for each steer.

## 3. Increasing Problems with Cattle Lameness and Mobility in Grain-Fed Steers and Heifers

The two recent studies discussed in the introduction clearly showed that lameness has greatly increased. The first study was conducted during 2021 and 2022 in five very large plants [1]. The five plants had chain speeds that varied from 150 to 390 cattle per hour [1]. They were located in the western, midwestern, and southwestern regions of the U.S. [1]. Eight percent of the cattle were lame [1]. The cattle were 89% Bos taurus and 71% graded USDA choice. In the second study, 32% of the fed cattle were lame [2]. Ninety-seven percent of the cattle were Bos taurus and 96% were under 30 months of age. Lameness was scored on a four-point scale.

(1)Normal walking;(2)Slight limp, keeps up with normally walking cattle;(3)Obvious limp, lags behind normally walking cattle;(4)Extremely reluctant to move. Adapted from Davis et al. (2024) [1].

Most of the cattle had a lameness score of two. Minor lameness compromises animal welfare. Twenty years ago, I visited all of the plants where this study was conducted. The incidence of lameness during truck unloading or when the cattle were moved in the alleys was almost zero. My evidence is anecdotal, but it is important to report it. Data collected at three bison slaughter plants in 2021 and 2022 indicated that only 0.56% of the bison had an abnormal mobility score [30]. A very recent study carried out in 2024 showed that the distance grain-fed steers and heifers traveled to the plant increased the probability of lameness [1]. Twenty years ago, these same five plants all existed, and the author observed that the feedlots were the same distance away. The average distance was 155.4 km. Years ago, the transport distance had no effect on these cattle because there was almost no lameness. Possibly, foot and leg abnormalities may be a factor in the increased occurrence of lameness in grain-fed steers and heifers hauled longer distances.

### 3.1. Increases in the Percentage of Grain-Fed Steers and Heifers with Hoof Abnormalities

During the last ten years, I have observed an increasing incidence of cattle with abnormal hooves. The most common abnormality is corkscrew hooves, in which the tips of the claws cross. It can range from a slight inward curl of the claws to completely crossed claws [31]. Cattle that develop corkscrew claws should be culled [31]. Both my own observations at slaughter plants and reports from industry indicate that corkscrew claws appear in some groups of cattle and not in others [31]. In 2023, I observed groups of grain-fed Angus steers at a large slaughter plant. In one pen of fed cattle, most of the animals had slightly crossed claws. My observations also indicated that many Angus x Holstein-cross, grain-fed steers also had many crossed claws. Melinda McCall, (personal communication, 22 February 2024) a cattle veterinarian based in Virginia, has recently observed an increase in the total cases of twisted claws. She sent me a photo of a four-month-old, grass-fed Angus heifer with severely crossed claws. In 2024, she stated that a single trait’s selection, for a high weaning weight, may be contributing to making this problem worse. Scoring tools for hooves and legs are available [32,33]. It is important not to confuse corkscrew claw with laminitis [31].

### 3.2. How to Reduce Lameness in Beef Cattle

The American Angus Association has recognized the importance of cattle having correct hoof and leg structures. In 2019, they developed a foot and leg structure EPD [33,34,35]. Research clearly shows that hoof and leg conformation ranges from less- to moderately heritable [34,35]. Breeders need to cull bulls and cows with poor hoof or leg structures. There also needs to be an increase in education for producers. 

### 3.3. Other Factors That Have Contributed to Lameness and Heart Issues

Other factors that may have contributed to these problems is that grain-fed steers and heifers are now heavier at a younger age. Carcass weights have also greatly increased [36]. Cattle have also become taller and wider. Since 2021, I have seen increasing problems with cattle getting stuck in the single-file leadup race at the beef plants. A 76 cm wide concrete race was standard for the industry for over 40 years. Today, a few huge steers are getting stuck in it. In 2023, one plant that specializes in large grain-fed feedlot cattle had to replace their restrainer system with one that was 15 cm wider. Since 1970, the average cattle live weights of grain-fed steers and heifers have increased by 30% [37]. Between 2023 and 2024, carcass weight has increased by 8.2 kg [37].

## 4. Ways to Improve Welfare

There are two approaches for improving welfare. I recommend breeding cattle that would have fewer heart and foot abnormalities. The installation of expensive equipment to handle animals that have abnormalities is not the right approach. The industry needs to produce cattle that are more mobile and easier to handle. A possible reason why heart and leg problems have become more prevalent may be due to what I call “bad becoming normal”. These problems slowly became worse, and many people did not recognize that lameness and heart problems were gradually increasing. Studies have shown that dairy producers will greatly under-estimate the percentage of lame cows [38,39]. It is really important to measure lameness, to prevent “bad becoming normal”. Dairy producers became so accustomed to looking at lame cows that they no longer perceived it as a problem. Young people entering the industry today may not know that 8% lame grain-fed steers and heifers is not normal. 

## 5. Grain-Fed Steers and Heifers Are Now Heavier at a Younger Age

Another factor that may have contributed to increased heart and foot problems in young, fed, grain-fed steers and heifers is that they are now heavier at a younger age. I am also concerned about pushing the animal’s biology to the point where it starts to break down. There are many other examples where breeders have over-selected for either appearance or production traits that have compromised animal welfare. Some examples are excessively flattened faces in bulldogs [40] and osteoporosis in laying hens [41]. Dogs with extreme brachycephalic short snouts may have breathing problems. Osteoporosis in hens may make them more prone to keel bone fracture. In Germany, this is called Qualzucht or extreme breeding or torture breeding [42]. Should animals “be genetically adapted to possibly unfavorable husbandry conditions or whether the husbandry conditions should be adapted to the animals” [42]. Increases in congestive heart failure and poor hoof conformation are defects that compromise welfare under a variety of husbandry conditions. They both cause severe welfare issues. In both pets and food animals, breeders should select animals that are functionally normal.

## 6. Other Causes of Congestive Heart Failure and Lameness

This section contains a discussion of the other causes of both congestive heart failure and lameness that were not covered in the main part of this paper. Another cause of heart failure is traumatic pericarditis due to a foreign body penetrating the reticulum [43]. This study was conducted in Egypt and was recommended by a reviewer. In this study, metal and other foreign bodies that penetrated the reticulum were associated with heart problems. Feed in both U.S. or Canadian feedlots would be less likely to contain metal or other foreign bodies that would injure cattle. Feed mills use magnets to remove metal. A study published in 1990 showed that feeding a high-energy ration increased laminitis in young, under-14-month-old calves [44]. Observations by a veterinarian in 1981 also indicated that an acidotic ration increased hoof problems [45]. A more recent review also discussed laminitis [46]. Mycoplasma bovis infection and digital dermatitis are also associated with lameness [47,48]. Only 22% of the cattle diagnosed with digital dermatitis were lame [49]. A survey on lameness in Alberta, Canada, feedlots indicated that there were big differences between the feedlots [49]. Lameness varied from 1.3% to 46% of the cattle [50]. It is likely that many severely lame cattle that were treated in feedlot hospitals did not go to slaughter with their penmates. A recent review, published in 2024, showed that 47.83% of the lame cattle were sold as railers or realizers [50]. They were sold for salvage slaughter. Many of the cattle with digital dermatitis were also culled early [51]. Necrosis syndrome often occurred early in the feeding period [52]. Anecdotal reports from people in the industry, and one of the reviewers, indicated that it may be related to injuries during the handling of wilder cattle. It is likely that many of these cattle were culled early in the feeding period.

The two surveys at the fed beef slaughter plants both revealed a small percentage of severely lame cattle [1,2]. It is likely that these more severely lame cattle may have the conditions that have been discussed in this section. No feet were examined in the surveys. Many severely lame cattle probably were sold as realizers or euthanized. The large number of cattle with mild lameness arriving at fed slaughter plants is a relatively new problem.

## 7. Conclusions 

Both hoof abnormalities and congestive heart failure have recently increased in grain-fed steers and heifers. This may be associated with genetic selection for economically important meat traits. Over the last ten years, this problem has gradually become worse. It happened slowly so many people did not notice that it was increasing. Cattle breeders need to select animals that do not have these abnormalities, because they severely compromise animal welfare.

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
