# Peer review of "Problems with Congestive Heart Failure and Lameness That Have Increased in Grain-Fed Steers and Heifers"

_animals, 2024, doi:10.3390/ani14192824_

Round 1

Reviewer 1 Report

Comments and Suggestions for Authors

Thank you very much for this interesting commentary. The connection between increasing health issues regarding lameness and heart failure, along with the need for genetic selection, definitely deserves closer consideration.

Hier ist die Übersetzung ins Englische:

"I recommend considering the following points for improvement:"

Abstract, lines 18 to 19: it would be best to replace the personal observation with documented and published concrete figures ('almost zero') from the specified period.

Keywords: I would recommend to add congestive heart failure and remove brisket disease

Text: The numbering is confusing; please organize the individual paragraphs with corresponding headings, without numbering.

Introduction: Lines 26–37: The first sentences refer to both health issues being discussed and should therefore come first. Afterward, subheadings (Lameness / Congestive Heart Failure) can be used.

From line 37: Subheading: Lameness

Lines 40–42: Please remove "To put these...facilities and."

The following sentence part "I (also) developed a simple scoring system for assessing cattle handling at THESE slaughter plants" should be placed in line 37 after "and in slaughter plants."

Lines 45 to 51: Here, I recommend deleting the sentences up to "... handling practices quickly improved" – there are many repetitions of the same statements, or they seem irrelevant.

Lines 54–72: "The penalty…or a noisy truck." Please condense this part. The content of these lines repeats what has already been said. While it is generally interesting, it strays into a narrative tone.

Lines 72 (starting from: "In the next section…") – 74: I would like to recommend deleting

Line 76: subtitle: Congestive Heart Failure

From line 77: Congestive heart failure can have various causes and also different symptoms. One symptom is swelling of the brisket, but it’s not the only one. The paragraph should start with a brief definition (e.g., "congestive heart failure describes the end-stage of a cardiac disease with an increasing hydrostatic pressure") followed by a description of the different symptoms (e.g. Weakness, intolerance to exercise, jugular vein distension, swelling/edema of the brisket). Among these, swelling of the brisket can be mentioned.

Line 82 and following: In addition to high altitude/mountain sickness, other very important causes should also be addressed. The most common one seems to be traumatic pericarditis due to foreign body penetration through the reticulum. Please refer to: M. Abd El Raouf et al., "Congestive heart failure in cattle: etiology, clinical and ultrasonographic findings in 67 cases," Vet World 2020, 1145-1152.

Line 95: please remove the numbering

From line 96: This is the place to discuss possible genetic causes in detail.

I would recommend omitting the authors' names from the text line 103/110 , as they are presented in the citation (e.g., Line 103: Isabella Kukor, line 110 Justin Buchanan and colleagues).

Suggested revision (Kukor´s study)

"In 2021, a study showed that 34% of the cattle slaughtered from Colorado feedlots exhibited signs of heart swelling. (4)." 

And here, the sentence from lines 122-123 should be included: 

„This study also showed that there was a sire effect on heart swelling on dairy steers (4) - supported by a personal communication from a feedlot manager in Colorado, who stated that he had traced cattle that died late in the feeding period back to a single Angus sire.“

Therefore, lines 120 to 126 could be removed.

Line 127: please remove numbering

Line 128-131: repetition, not necessary. Additionally, much of the following text repeats what has already been mentioned in the Introduction. Please revise it accordingly - the focus here should be on presenting possible genetic issues.

Line 152 and 166: please remove numbering

Line 153-154: are there any published figures on this?

Line 164: please use „Laminitis“ instead of founder - or better: claw horn disruption.

 Also, please consider that digital dermatitis in beef cattle is increasing significantly and can cause hoof problems/lameness (e.g. JA Cortes et al., Risk factors of digital dermatitis in feedlot cattle, Transl Anim Sci 2021) . The causes are partly related to housing conditions, but also, of course, to genetics.

Line 167-173 This should be included in the discussion as a potential improvement, which has already been applied to the Angus breed with considerable success!

Line174: please remove numbering. 

175 - 183: This paragraph discusses another possible cause for increasing lameness and should, therefore, be placed in the text after the section on hoof abnormalities.

Line 184: please remove numbering and use „Discussion“

First, discuss whether the causes of congestive heart failure might also be related to the increasing presence of foreign bodies in the feed (e.g., from harvesting processes) and how addressing these causes could be beneficial. Next, emphasize that genetic selection can have a significant positive impact, as demonstrated with Angus cattle (use the section from lines 167-173). Then, revisit the discussion on the weight development of the animals (lines 199-201) and consider introducing and briefly defining the term 'suffering breeding' (Qualzucht).

For the discussion, I would also like to consider whether animals should actually be genetically adapted to possibly unfavorable husbandry conditions or whether the husbandry conditions (see line 175 to 183 and „risk factors for lameness“) should be adapted to the animals. Both aspects probably need to move towards each other.

Reviewer 2 Report

Comments and Suggestions for Authors

1. Professor Grandin: While there can be no argument with your conclusions, the reasons behind those conclusions are complicated.

a. In recent years more Angus and Angus X animals have been placed in NA feedyards. They come with CHF DNA

b. CHF is also more important as a complication of BRD than it was in the past. Pathogens like Mycoplasma bovis are easy for stock attendants to overlook, the result being CHF can be an outcome of longstanding M bovis caused BRD

2. As our cow herd in NA decreases in number, carcass weights have been increasing to maintain beef in the supermarkets. Thus cattle, as a rule have greater DoF, thus increasing the risk of "Rumen acidosis induced" Laminitis.  No surprise more animals presented to the abattoir have various degrees of lameness.

3. In addition, as ranchers move breeding to lessen dystocias, the trend to later calving on range has developed. Cattle raised away from human contact tend to develop a "wilder" disposition. And, those cattle are associated with an observed development to the "Toe-Tip Necrosis Syndrome", because of excessive hoof wear and whiteline injury. They too are very lame, but likely never get to the abattoir.

Reviewer 3 Report

Comments and Suggestions for Authors

The author comments on the increase in the incidence of lameness and right-sided heart failure in feedlot cattle. The commentary is compelling and should give pause to the cattle industry. Some attention to the timeline and statements in the referenced studies would improve the arguments.

Line 28: The cited paper states that 7.8% of cattle fall into the category “exhibits minor stiffness, keeps up with normal cattle “, while 0.3% and 0.002% have more major degree of lameness. It would be desirable to have no lame cattle at slaughter, but some qualification of the 8% lame might be indicated – same for reference 2. – I see that later on in the manuscript there is mention of the scoring scheme. I would suggest that some verbiage such as “some degree of lameness” is used in this section, rather than the word lame, which may conjure up a different image than what is the actual reality.

Line 79: this should be “blood flow” instead of “flood flow”.

Line 81: instead of “arteries” I suggest “pulmonary artery”.

Line 84: I do not find the statement that heart disease in cattle has a low incidence in the study by Buczinscki et al. The authors in the Buczinski paper state that “Bovine cardiac disease has received little clinical attention because an early diagnosis is not frequently made and most conditions progress to heart failure, which has a poor prognosis.” The study is a retrospective case study of 47 cattle suffering from heart disease without clinical signs of heart failure. The study did not try to estimate the incidence of heart disease.

In fact, the study by Neary et al. reference # 25 found increases in the risk of right-sided heart failure in Canadian feedlots cattle between the years 2000 – 2012, indicating that the increase in risk might have occurred earlier than 2015, as you  also mentioned.

Reference 26: The third author’s name is Grotelueschen, not Grolelueschen

Line 89: Although the study was published in 2019, the data in the study is from 2009 -2010.

Line 109: This statement is not correct. The referenced study says that 29% of the cases were homozygous for the risk alleles for  both ARRDC3 and NFIA, which is a different statement from the commentary that says that 29% of cattle that are homozygous for the ARRDC3 allele get heart disease.

Line 112: The feedlot was in Idaho, the elevation mentioned in the reference was 756 m. The cattle were sourced from the West and Pacific Northwest cow-calf and beef-on dairy operations.

Line 113: This sentence could be fleshed out more to underscore its significance: because the authors found positive correlations of heart scores with growth traits and feed intake, selecting for these traits may have inadvertently selected for cattle prone to CHF as well.

Reference# 20: This study was published in 2008 not 2018. I suspect that no reference manager was used for this manuscript due to minor typos in the references. I highly recommend use of a reference manager such as Endnote or Mendeley to avoid such errors.

Line 117: The statement should be modified to represent the findings in the study: The probability of noninfectous heart disease increased with increasing days on feed from 1.5% to 2.87% until 326 days on feed. With more DOF there wasn’t an additional increase in these deaths.

Line 124: “beef-on-dairy “ with the hyphens

Lines 128 and 140: I do appreciate that the author does not recall any lameness in cattle at that time. It would be even more compelling if there were any published data on lameness in slaughter cattle from that time period. One could argue that the absence of such studies might be an indicator that it wasn’t an issue at the time. I’m not sure, but the author must admit that we are talking about anecdotal evidence in this case.

Line 142: “Another”? where is the first indicator? Maybe just say “A possible indicator…”

Line 145: the study states that 0.56 % of bison scored an abnormal mobility score under the paragraph “Benchmarking” in the results section.

Line 148: This statements seems unfounded: in the cited reference # 31 the authors state that the continual consolidation of both feedlot operations and processing plants has increased the distances that animals are transported to reach these facilities. Additionally, economic drivers may also increase transportation distance. Increased distance traveled was associated with increased odds of mobility impairment in cattle. Clearly, an issue such as lameness is complex because there is more than one cause for lameness in cattle, but the author’s observation that previously cattle were not lame when arriving at the plant and therefore transportation cannot be blamed could be applied to any other potential cause as well.

Line 162: is there a reference for Dr. McCall’s statement?

References # 32 and 33 seem to be identical.

Comments on the Quality of English Language

There are some minor issues with grammar. It would be helpful if one more person familiar with the subject matter, such as a grad student, would read and edit the manuscript.

Round 2

Reviewer 3 Report

Comments and Suggestions for Authors

Thank you for your responses. I don't have any further comments on this manuscript.

Comments on the Quality of English Language

A few typos remain. Please give it one more good read-through before publication. E.g. the author of reference #26 still has a typo. The correct spelling is Grotelueschen.

Author Response

I have revised my manuscript Animals 3201994 per the comments by Reviewer 3.